# Real-space observation of ergodicity transitions in artificial spin ice

Michael Saccone [1] ✉, Francesco Caravelli [1], Kevin Hofhuis[2,3], Scott Dhuey[4], Andreas Scholl[5], Cristiano Nisoli[1] & Alan Farhan [6] ✉

Ever since its introduction by Ludwig Boltzmann, the ergodic hypothesis became a cornerstone analytical concept of equilibrium thermodynamics and complex dynamic processes. Examples of its relevance range from modeling decision-making processes in brain science to economic predictions. In condensed matter physics, ergodicity remains a concept largely investigated via theoretical and computational models. Here, we demonstrate the direct real-space observation of ergodicity transitions in a vertex-frustrated artificial spin ice. Using synchrotron-based photoemission electron microscopy we record thermally-driven moment fluctuations as a function of temperature, allowing us to directly observe transitions between ergodicity-breaking dynamics to system freezing, standing in contrast to simple trends observed for the temperature-dependent vertex populations, all while the entropy features arise as a function of temperature. These results highlight how a geometrically frustrated system, with thermodynamics strictly adhering to local ice-rule constraints, runs back-and-forth through periods of ergodicity-breaking dynamics. Ergodicity breaking and the emergence of memory is important for emergent computation, particularly in physical reservoir computing. Our work serves as further evidence of how fundamental laws of thermodynamics can be experimentally explored via real-space imaging.

Artificial spin ice is a term that summarizes a range of nanoscale model magnetic systems that feature various forms of geometrical frustration[1]. They consist of single-domain Ising-, Potts- and XY-type nanomagnets lithographically defined onto various two-dimensional geometries[2–6]. While initial realizations mimicked the geometrical frustration in naturally occurring pyrochlore spin ice[7–10], advancement in fabrication and characterization allowed for creative system designs and the realization of a variety of artificial frustrated systems to exhibit emergent phenomena often not seen in natural materials. These include emergent magnetic charge screening[3,4], emergent reduced dimensionality[11], spin glass transitions[12], or the direct observation of phase transitions[13]. Artificial spin ice systems, combined with appropriate magnetic imaging techniques, are now situated as ideal platforms to generate and visualize emergent phenomena and fundamental laws and concepts of thermodynamics that would otherwise rely on simplified theoretical models.

For example, the ergodic hypothesis introduced by Boltzmann presupposes that thermodynamic systems explore all allowed states of given energy in proportion weighted by their multiplicity, bringing averages over repeated ensembles quickly into agreement with averages over time. However, exceptions are predicted to occur as ergodicity tends to be broken around phase transitions, freezing

[1]Center for Nonlinear Studies and Theoretical Division, Los Alamos National Laboratory, Los Alamos, NM 87545, USA. [2]Laboratory for Mesoscopic Systems, Department of Materials, ETH Zurich, 8093 Zurich, Switzerland. [3]Laboratory for Multiscale Materials Experiments (LMX), Paul Scherrer Institute, 5232 Villigen PSI, Switzerland. [4]Molecular Foundry, Lawrence Berkeley National Laboratory, One Cyclotron Road, Berkeley, CA 94720, USA. [5]Advanced Light Source, Lawrence Berkeley National Laboratory, One Cyclotron Road, Berkeley, CA 94720, USA. [6]Department of Physics, Baylor University, Waco, TX 76798, USA. ✉e-mail: msaccone@lanl.gov; alan_farhan@baylor.edu

transitions, or spin glass phases[14]. Furthermore, local constraints in quantum systems can also lead to ergodicity-breaking dynamics[15]. Geometrically frustrated systems such as artificial spin ice exhibit frustration and strong local constraints in form of a strict adherence to the so-called ice-rules[16] while exploring their energy landscape via thermally-driven moment fluctuations[17]. Violations of the ice-rule come with an energy cost and the emergence of topological defects, such as emergent magnetic monopoles[8,9].

In this work, we aim to directly visualize ergodicity transitions in a vertex-frustrated artificial spin ice, which we dub the Apamea lattice (Fig. 1a, b), as it features similarities to Roman ornaments recovered in the region surrounding the city of Apamea. The lattice features both four-nanomagnet and three-nanomagnet vertices as seen in previous studies of the decimated square lattice[11]. These vertex-frustrated systems such as the Tetris lattice[11], the aforementioned Shakti lattice[4], or the so-called Cairo- and Santa Fe lattices[18,19] have proven particularly interesting, as they feature a strict adherence to ice-rule constraints at the four- and three-nanomagnet vertices but not all vertices can be placed simultaneously in the lowest energy configuration.

## Results

### The dipolar Apamea spin ice

The Apamea lattice organizes three-nanomagnet vertices into square-shaped windows, which are then connected via four-nanomagnet vertices and two-nanomagnets vertices (Fig. 1a). Moment configurations (see example in Fig. 1b) can then be characterized by looking at the so-called vertex types, which are listed in Fig. 1c with increasing dipolar energies. From an emergent magnetic charge representation[3,8,10], four-nanomagnet vertices obeying the ice-rule (Type I and Type II vertices) will feature zero magnetic charge at the vertex, while a Type III and Type IV vertices will exhibit non-zero magnetic charge defects (Fig. 1c). In the three-nanomagnet vertices, the ice-rule obeying vertices are energetically split into Type A and Type B vertices, with Type B being higher in energy due to the asymmetry in interactions between collinear and perpendicular nanomagnets. These ice-rule vertices of odd coordination are necessarily charged, with magnetic charge $Q = \pm q$, $q$ representing the charge of a single nanomagnet pointing into a vertex. These types of four- and three-nanomagnet vertices are expected to show a strong, if not strict, adherence to their respective ice-rules[4,11,18]. In other words, we are

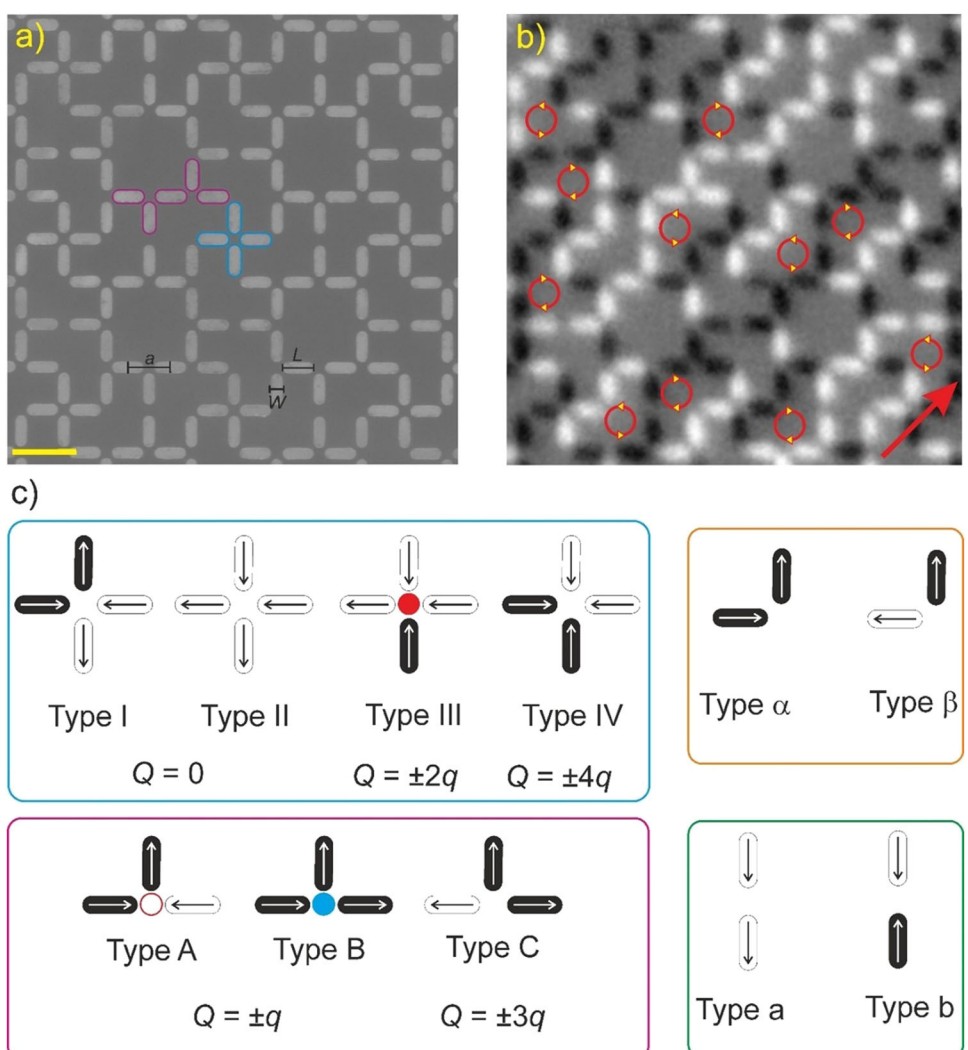

**Fig. 1 | Apamea lattice. a** Scanning electron microscopy image of the Apamea lattice consisting of nanomagnets with lengths $L = 360$ nm, width $W = 120$ nm, and thickness $d = 2.6$ nm with a lattice parameter $a = 500$ nm. The yellow scale bar indicates a length of 600 nm. **b** XMCD image of low-energy moment configuration recorded after thermal annealing. While ground state domains emerge, we see sporadic clockwise and anti-clockwise moment loops forming on the irregular hexagonal plaquettes. **c** Vertex types at two-, three-, and four-nanomagnet vertices listed with increasing dipolar energy. For example, the order is from 1 through 4, A through B, etc. Magnetic charges at each vertex, $Q$, are listed in terms of multiples of the dipole's fundamental magnetic charge $q$. Positive $Q$ values are depicted as red and negative values are blue.

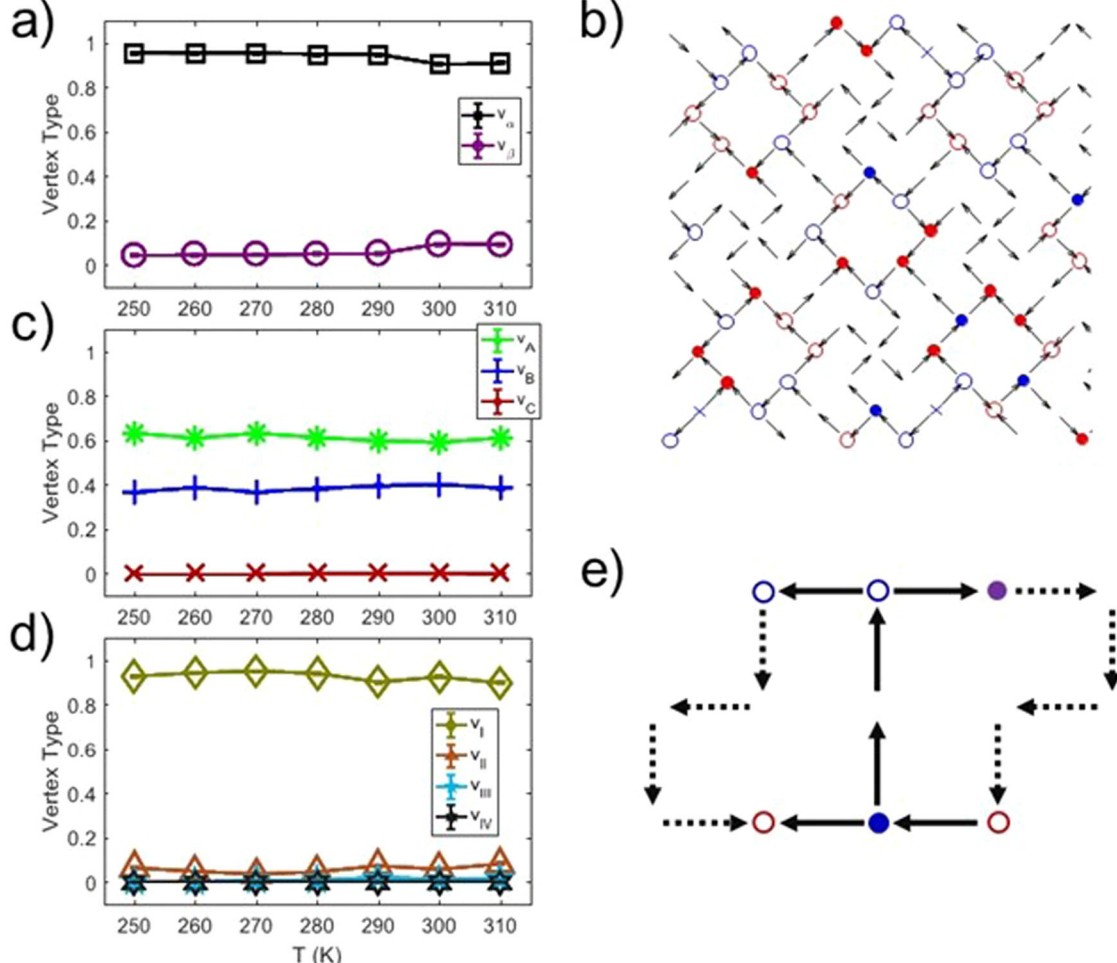

**Fig. 2 | Thermodynamics of the dipolar Apamea lattice. a, c, d** Vertex-type populations plotted as a function of temperature for two-, three-, and four-nanomagnet vertices, respectively. Error bars, calculated from the standard deviation of the mean over every XMCD frame, are present but smaller than the size of the markers. **b** Magnetic moment and charge configurations extracted from a single frame within an XCMD image sequence recorded at 290 K. Red shapes are positive charges, blue are negative, open circles are Type A vertices, higher energy Type B vertices are shown by closed circles, and pluses denote Type b vertices.

**e** Schematic of a geometrically useful pair of plaquettes in a low-energy configuration. Under the conditions that only Type I and Type α vertices may exist, this pair of plaquettes is impossible to fill with entirely Type A vertices. Specifically, at least two of six of the coordination 3 vertices must be Type B. One of the Type B vertices is colored purple to denote that its sign will be determined by an undrawn spin. The dotted arrows indicate spins which are shared by paired plaquettes, making the total number of spins per paired plaquette 10.

studying a magnetically frustrated system with strong constraints on local dynamics, in the form of ice-rule obeying dynamics. Similar to other vertex-frustrated systems[11,18,20], the Apamea lattice exhibits a predicted ground state when considering a dominance of nearest-neighbor couplings (see Supplementary Fig. 1), but which requires a staggered presence (25% population) of so-called *unhappy* Type B vertices that, despite fulfilling the ice-rule, are higher in energy than the *happy* Type A vertices (Fig. 1c).

### Temperature-dependent thermal fluctuations

With this a priori knowledge about the system, we now turn our focus to our temperature-dependent magnetic imaging experiments[2,3]. For this, we patterned Ising-type nanomagnets, arranging them onto an Apamea lattice (Fig. 1a). The total system comprises over 28000 nanomagnets which, though not a thermodynamic limit in that it does not approach Avogadro's number, is consistent with system sizes of nanomagnets that avoid significant edge and boundary effects, as previously reported in literature[3,8,9,12,21–23]. Following sample fabrication, the sample is kept at room temperature for several weeks before being transferred into the photoemission electron microscope

(PEEM)[24] for magnetic imaging. Employing x-ray magnetic circular dichroism (XMCD) at the Fe L3 edge[25] (see "Methods" section), we are able to record XMCD image sequences at various temperatures with a temporal resolution of roughly 10 s per image (see Supplementary Movies 1 and 2).

Looking at the recorded XCMD sequences, we first count vertex-type populations as a function of temperature (Fig. 2a, c, d). While the plots appear unspectacular on a first glance, showing no dramatic change with temperature, they reveal two important conclusions. The unhappy Type B vertices maintain a fraction close to 0.4, higher than the predicted nearest-neighbor ground state fraction of 0.25 (see Supplementary Material), while the rest of three-nanomagnet vertices occupy Type A vertices. Type C are never observed, as they are too energetically unfavorable (Fig. 2c). Second, Type I vertices dominate the four-nanomagnet vertex sites at all temperatures, with over a 0.9 fraction throughout all accessible temperature ranges. Type II vertices fluctuate consistently around 0.1 with small error bars (Fig. 2d), as Type II and Type I vertices keep converting between each other via Type III creation and annihilation, in analogy to square ice dynamics[26]. The strict constraint within the three-nanomagnet vertices

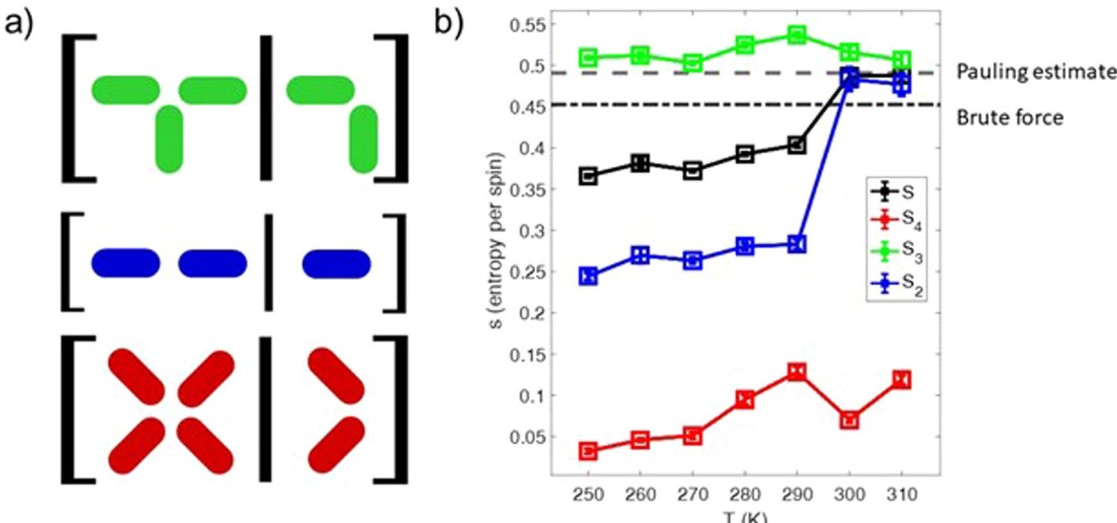

**Fig. 3 | Conditional entropy as a function of temperature. a** Conditional schemes used to estimate the entropy of different subsets of the system. For example, the green coordination three vertices have their entropy estimated by first determining what the entropy of all three spins is when information from the first two is known. **b** Entropies plotted as a function of temperature compared to the theoretical predictions. The total bitwise entropy, $s$, is the average of the three types of entropy weighted for their frequency of occurrence in the system. This total entropy quickly drops beneath both theoretical estimates. These entropies were calculated with every XMCD image and their error bars result from standard deviation of the mean over the images at each temperature.

only allowing back-and-forth conversions between Type A and Type B vertices serves as a significant barrier for the system to find an easy pathway towards long-range ground state ordering. In other words, the system is stuck in a low-energy configuration that does not pass through higher energy states to find the ground state, for both charge- and moment degrees of freedom (Fig. 2b).

Interestingly, while vertices with two and four islands are very close to their ground state, three island vertices are excited to their Type B state in much higher frequencies than the low-energy config- uration reported in Supplementary Material. Note, however, that the long-range nature of the dipolar interaction would favor the closure of fluxes around L-Shaped plaquettes. Closing their flux around square and irregular hexagonal polygons formed by the nanomagnets, here- after referred to as plaquettes, decreases long-range dipolar energetics as highlighted in Figs. 1b or 2e. On the other hand, maximizing Type A vertices works against moment loop formations in the plaquettes (Fig. 2e). In other words, the state that minimizes the number of unhappy vertices might not be the real ground state of the system. To gain a deeper understanding, we turn to measures of entropy to characterize this behavior.

### Direct entropy determination
We extract an upper bound on the observed entropy from moment configurations recorded as a function of temperature using local conditional entropy[2,27] (see "Methods" section). To do so, following ref. 24, we consider subsets of the lattice, coordination two, three, and four island vertices (Fig. 3a), and ask the question: How much infor- mation is hidden by only observing a part of this subset? If the rest of the microstate is determined by only the part of the subset with perfect probability, the entropy per bit is zero, but if the rest of the subset is entirely random, the entropy per bit is one. The details of the inter- mediate probabilities are determined by Bayes' theorem and basic information theory[2,27] (see "Methods" section). Calculating the entropy for each temperature and coordination of vertex, as well as the appropriate weighted average to determine the total entropy, we see that the entropy bound generally decreases with decreasing tempera- ture as expected (Fig. 3b), but with a stark jump downward between 300 and 290 K for the coordination two and four vertices while the bound from the coordination three vertices remains nearly constant.

As a first step towards understanding this observed entropy, we compute theoretical thresholds for comparison. Consider a model system that obeys the following rules. In agreement with energetic minimization and the visible spin configurations, we assume that four- nanomagnet and two-nanomagnet vertices are in their Type I and Type $\alpha$ configurations respectively. The number of ways in which Type B vertices may be configured determines the entropy of this model via counting the microstates permitted. We assume an average of two Type B vertices per paired plaquettes (Fig. 2e). Counting this is non- trivial as four of the six Type B sites per paired plaquettes are shared with neighboring sets of paired plaquettes, and therefore the multi- plicity of configurations depends on the configuration of their neigh- bors. The first estimate and highest bound of this counting is the so- called Pauling upper-bound, where the dependence on neighbors is simply ignored, while the second threshold and stricter upper-bound is calculated by a brute-force placement of Type B vertices as shown in Fig. 2e and Supplementary Fig. 2 and in the "Methods" section.

These bounds are compared to the experimental bitwise entropy[2,27], $s$ (see "Methods" section), which is extracted by consider- ing various block types (Fig. 3a), weighted for their frequency of occurrence, and plotted as a function of temperature (Fig. 3b). Inter- estingly, the entropy bound from four-nanomagnets vertex sites is close to zero, dropping below 0.05 entropy per bit at 250 K (red squares in Fig. 3b, described in Supplementary Fig. 2), which speaks to the high level of Type I vertex ordering these sites. This is consistent with the proposed nearest-neighbor ground state (Supplementary Fig. 1) as many coordination two and all coordination four vertices may order independently. The situation changes dramatically when con- sidering nanomagnets involved in three-nanomagnets vertex sites, where entropy fluctuates around the Pauling estimate (green squares in Fig. 3b). This corresponds to the type A and B vertices occurring at similar rates at interchangeable locations. Essentially, moment con- figurations strictly adhere to the ice-rule constraints while long-range order is absent. This stands in contrast to the ordered Type I vertices at the four-nanomagnet vertices.

It is possible that the nearly immovable Type I and Type $\alpha$ ver- tices block ergodicity, but also that long-range order begins to pro- hibit some local configurations despite the vertex populations remaining about the same. We compare these two factors by

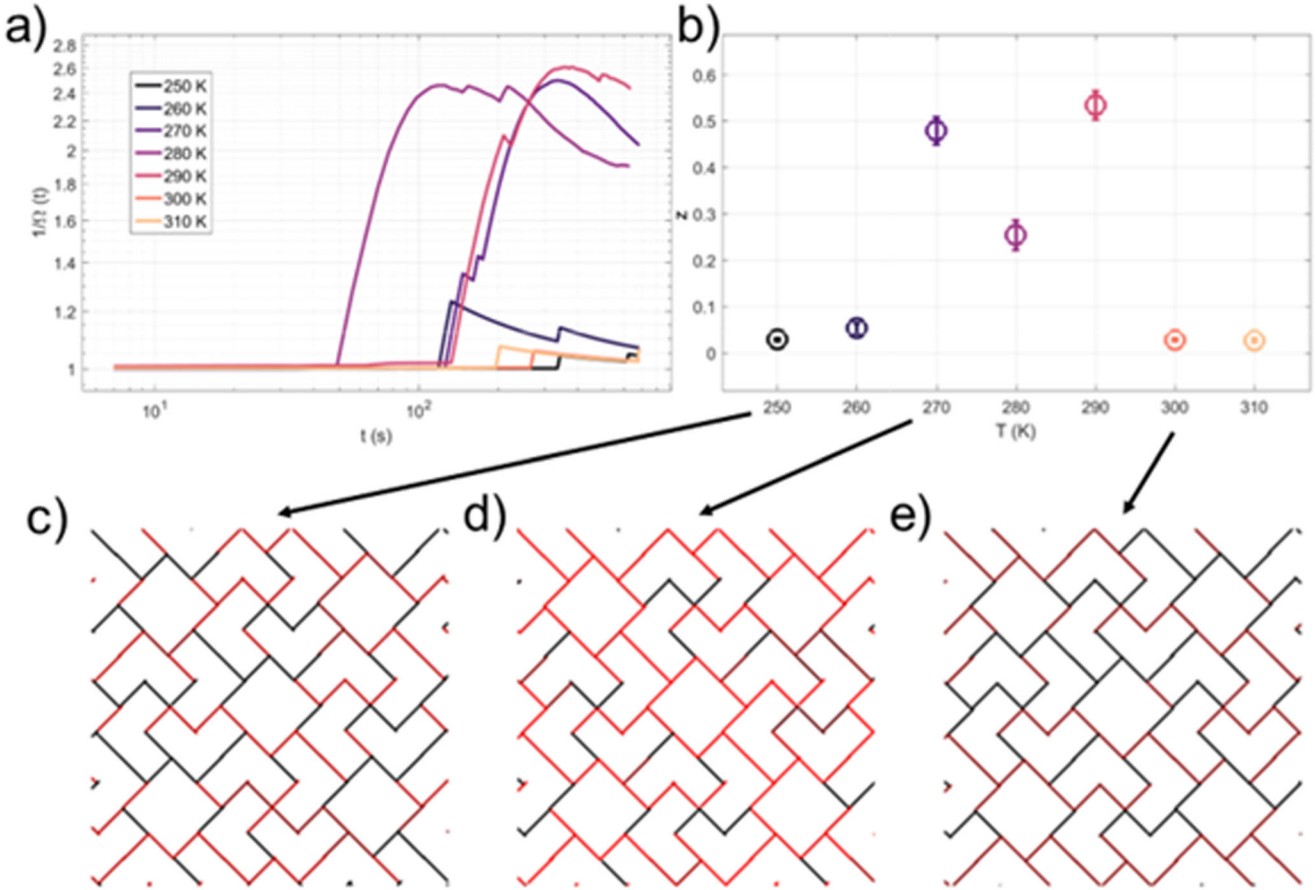

**Fig. 4 | Analyzing ergodicity via the stress metric. a** Inverse stress metric $\Omega^{-1}$(t) plotted for each temperature as a function of time on a log-log scale. **b** Exponent $z$ describing the stress metrics' decays extracted by ordinary linear least squares fits of the stress metrics. Error bars emerge from the standard error of fit. **c–e** Heatmaps depicting the frequency of nanoisland evolutions at (**c**) 250 K, (**d**) 270 K, and (**e**) 300 K. Nanoislands that are purely red fluctuate at ten times or more during the experiment duration while those that are purely black do not fluctuate at all.

calculating an appropriate measure of ergodicity from the time-resolved data.

## Stress metric and ergodicity

The general recipe to measure a dynamical system's ergodicity is to define an appropriate coordinate and see if it randomizes rapidly enough over time. In an ergodic system, the time average of a coordinate is equivalent to the spatial average of the coordinate given a large enough sample of system behavior. A stress metric[28], $\Omega(t)$, is defined as the difference between the spatial average and the time average as the sample is observed for time $t$[29] (Methods). This metric typically decays as a power law, $\Omega(t) \propto t^{-z}$, and ergodic systems decay with $z = 1$ while ergodicity breaking systems have values for $z$ between 1 and 0 due to their getting caught in a subset of all possible states. We calculate the stress metric of our system by considering the "spatial" average to be the Ising spin coordinates over each frame while the "temporal" average is the Ising coordinate average over all images successively captured (Methods). We calculated the stress metric at each time step of every recorded temperature (Fig. 4a) to extract the decay power $z$ via an ordinary least squares linear regression of the quantity $\log(\Omega(t))$ vs. the log of time, $\log(t)$, deriving the standard error of fitting parameters in the process, and plot its behavior over temperature (Fig. 4b). The value of $z$ fluctuates around zero at both high and low temperatures as the system appears to reject relaxation beyond certain regions of phase space. That is, the system quickly returns from brief fluctuations, seen as abrupt rises in the stress metric. Low temperature freezing is common in artificial spin ice as the fluctuation rate of permalloy diminishes, but the high temperature

systems remaining in the same basin is peculiar. Because the system is well annealed after over 4 weeks at 300 K, it is likely deep within a basin of attraction and limited in the moves it can make to escape.

By reaching an intermediate temperature, 270–290 K, the system fluctuates towards a new equilibrium ensemble. Since thermodynamic systems are forced towards free energy minima, and lowering the temperature shifts them, the system evolves anew to follow a new minimum. Lower temperature targets favor lower energy above multiplicity of states, pressing the Apamea lattice to seek long range order despite the kinetic barriers. This behavior hides in the fast relaxation of traditional materials, but the invisible is made visible in artificial nanomagnets.

The inverse stress metrics at the intermediate temperatures rise quickly then stabilize again, meaning that they quickly reach a weak ergodicity breaking state. The time average asymptotically differs from the spatial average, proof of an ergodicity-breaking transition. That is consistent with the measured fluctuation rates of individual islands. Figure 4c–e shows that fluctuation rates are generally higher for intermediate temperatures and there is a discrepancy of fluctuation rates between individual islands, seemingly due to both what quenched disorder they exhibit and what vertex coordination to which they belong.

Generally, coordination two and four vertices are less active and islands belonging to coordination three vertices are more active (consistent with their higher entropy), especially at intermediate temperatures. Regarded in combination with the drop in total entropy at 290 K and high coordination three entropy, we can conclude that the system at intermediate temperatures rearranges Type B vertices

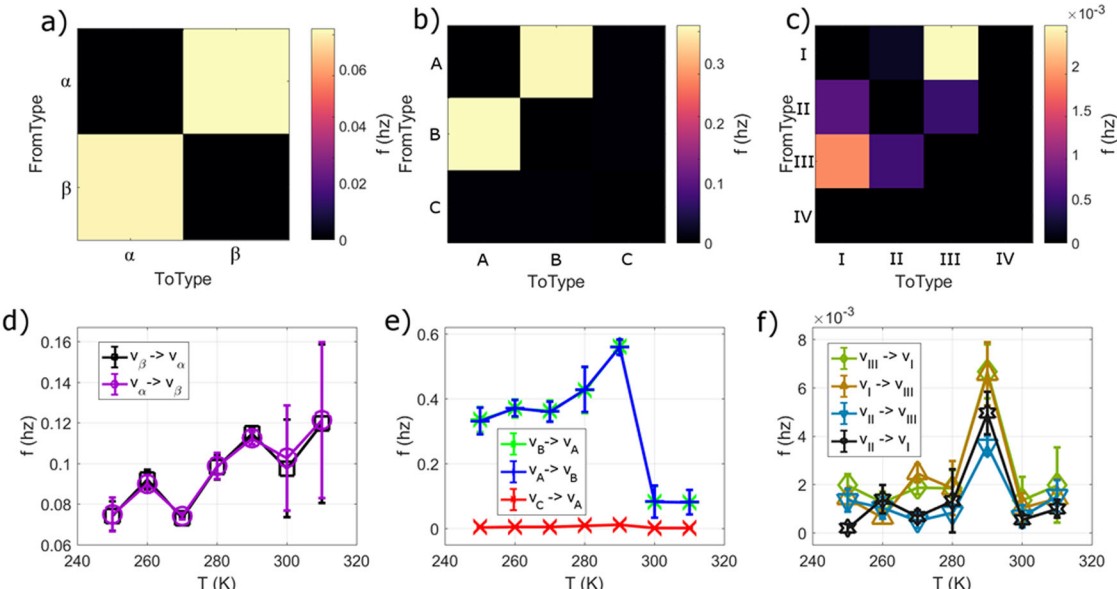

**Fig. 5 | Transition frequencies between vertex types. a–c** Heatmaps of transition frequencies from one vertex type to another at 270 K for **a** two-nanomagnet vertices, **b** three-nanomagnet vertices, and **c** four-nanomagnet vertices. **d–f** Select transition frequencies as a function of temperature for **d** two-nanomagnet vertices, **e** three-nanomagnet vertices, and **f** four-nanomagnet vertices. Error bars are the standard deviation of the mean when averaging the rates over four separate time intervals.

but is still dynamically constrained, a feature of weak ergodicity breaking. Looking at Fig. 4c–e, we can see that when the exponent is higher, the fluctuating spins represent a percolating sublattice, while when the exponent is closer to zero, only a non-percolating sublattice fluctuates. This is consistent with the behavior of the stress metric, from which we can conclude that subdiffusivity arises from the inability of the defects to completely explore the lattice.

To seek the microscopic mechanism of ergodicity breaking, we count the transition rates of vertex types from one to another (Fig. 5) at each temperature. A rate was calculated for four equal windows of time so that the error of the transition rates could be estimated via their standard deviation of the mean. Irrespective of temperature, the three island vertex transitions that cost no energy, $v_A \leftrightarrow v_B$, were most common at all temperatures but 300 K and 310 K ($f = 0.08 - 0.6 hz$, Fig. 5b, e), followed by $v_\alpha \leftrightarrow v_\beta$ conversions ($f = 0.7 - 0.12 hz$, and Fig. 5a, d) that dominate at higher temperatures, with infrequent transitions of the four island vertices ($f = 0.0005 - 0.008 hz$, Fig. 5c, f). Excursions to the higher energy vertices for three- and four-island vertices ($v_C$ and $v_{IIII}$ respectively) were essentially prohibited. Rates backwards and forwards between states were typically the same within error.

In the weak ergodicity breaking regime ($T = 270-290$ K), a couple of interesting kinetic pathways are present. Foremost, the rate of $v_A \leftrightarrow v_B$ and $v_I \leftrightarrow v_{III}$ transitions greatly increase, supporting the notion that energy equivalent fluctuations are occurring to reduce the entropy of the system. Additionally, there is the possibility of $v_{II}$ vertices becoming $v_I$ via short-lived Type III vertices. The increase in transitions involving the four-nanomagnet vertices becomes particularly strong at 290 K (Fig. 5f), before all such transitions dramatically fall in frequency and the system emerges out of the weak ergodicity-breaking regime at 300 K and above.

Overall, it appears that the increasing activation of transitions within the four-nanomagnet vertices together with an increase in transitions within two- and three-nanomagnet vertices, coupled with a strict ice-rule adherence at three-nanomagnet vertices is the main factor leading weak ergodicity breaking. Once fluctuations at four-nanomagnet vertices slow down, dynamics return to frozen behavior. Figure 6 features examples of both dynamical regimes, first at

$T = 290$ K (Fig. 6a), where transitions within the four-nanomagnets vertices occur regularly, while three-nanomagnet vertices keep fluctuating under strict ice-rule obedience. In contrast, at $T = 300$ K (Fig. 6b), transitions within four nanomagnets become far less frequent, corresponding to a return to frozen behavior.

## Discussion

Unlike natural magnets, artificial spin ice systems can allow for the direct visualizations and quantification of their degrees of freedom. We have used PEEM characterization to directly extract stress metrics, computed on the elementary degrees of freedom, to characterize ergodicity. We have also microscopically identified kinetic pathways for ergodicity transitions in the Apamea spin ice, which we have presented here for the first time.

This provides a route to explore system size and boundary effects[30], not only on vertex-populations, but on ergodicity itself. Despite a limited viewport into a system short of the thermodynamic limit, the nanomagnets failed to explore their more limited phase space. Other causes of ergodicity breaking must have been at fault. When combined with simulations, potential effects of intrinsic disorder[31] on the observed transitions can be characterized. Emerging characterization methods such as x-ray photon correlation spectroscopy (XPCS)[32] will be ideal to shed light into relaxation processes and the potentially glassy state the Apamea lattice settles into at lower temperatures, which might also be accessed indirectly using SQUID magnetometry[33]. Furthermore, our work will allow for direct comparisons to other complex systems in nature where ergodicity transitions are predicted to play a major role in the reorganization of the free energy landscape, from electron transfer processes in biological systems[34], neural networks[35], and fluctuations in quantum systems[15,36,37]. It is also important to stress that over the last few years device applications of spin ice materials have emerged, including collective computation[38,39] and in particular reservoir computing, a framework for machine learning prone to physical implementation[38,39]. Nanomagnetic islands with multiple magnetic states[38,39] have risen in interest because they enhanced the so-called computational memory capacity (CMP)[38,39]. Ergodicity breaking is indeed important in this respect, as memory arises in situations in which a physical system

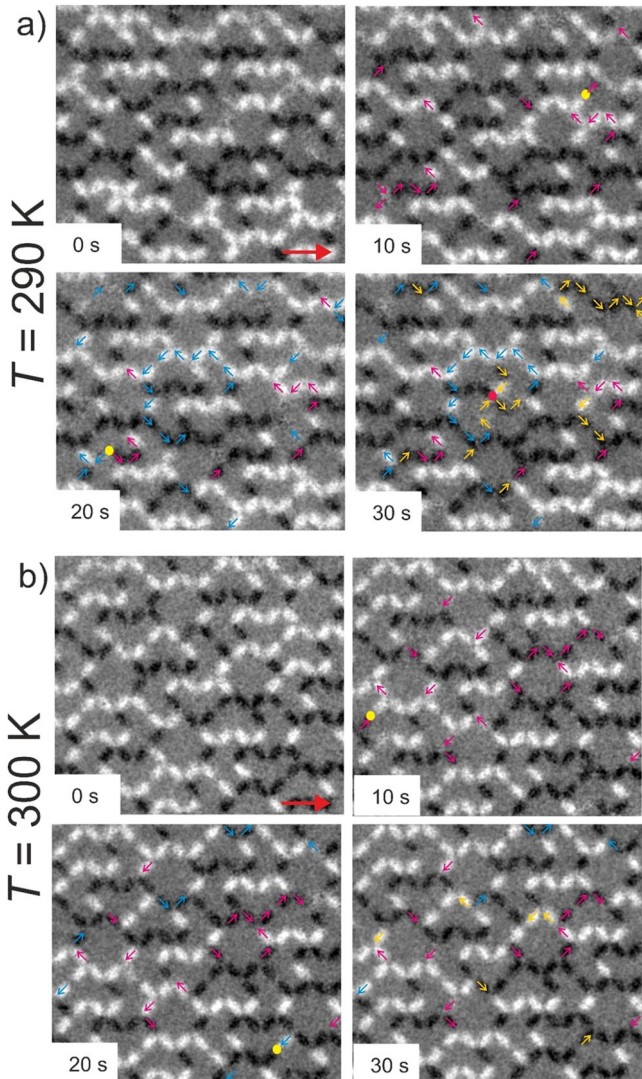

**Fig. 6 | Temporal evolution of moment reorientations in the dipolar Apamea lattice. a** XMCD image sequence (recorded at $T = 290$ K) highlighting vertex transitions within a weak ergodicity-breaking regime, which involves moment reorientations in all vertex types. Arrows of different colors (magenta, blue and yellow) indicate sequential changes of moment configurations at corresponding time-frames (10 s, 20 s, and 30 s). Dynamics within this regime involve increased frequencies of vertex-type transitions within all vertices, be it two-, three- and four-nanomagnet vertices. **b** XMCD image sequence (recorded at $T = 300$ K) highlighting vertex transitions within an ergodic regime, which largely excludes transitions within four-nanomagnet vertices.

maintains a proximity to its initial state. Our work is thus a first important milestone for the design of optimal nanomagnetic systems able to enhance the CMP in physical systems.

## Methods
### PEEM imaging
Magnetic imaging was performed at the PEEM3 beamline of the Advanced Light Source, Lawrence Berkeley National Lab[24]. We employ x-ray magnetic circular dichroism (XMCD) at the Fe L3 edge[25]. A typical XMCD image is a result of dividing two images, one obtained with circular left and the other with circular right polarized x-rays. The dark and bright contrast in an XMCD image is a direct measure of the relative orientation of the local magnetization with respect to polarization vector of the incoming x-rays. Moments pointing towards the incoming x-rays will appear dark, while moments opposing the

incoming x-ray direction will appear bright. To record XMCD image sequences, we set the exposure time at 1 s at each polarization, while the time needed for switching the polarization is about 4 s. As a result, 10 s are needed for an individual XMCD image. We recorded 70–100 images at each temperature set as 250 K, 260 K, 270 K, 280 K, 290 K, 300 K, and 310 K.

### Sample fabrication
Lift-off assisted electron beam lithography was used to generate dipolar Apamea lattices. A $1 \times 1$ cm$^2$ Silicon (100) substrate is first spin-coated with a 70 nm thick layer of polymethylmethacrylate (PMMA, 950k) resist. Then, a VISTEC VB300 e-beam writer is used to place Apamea lattice patterns onto the substrate. After the exposed resist layer is developed, a 2.6 nm thin Permalloy (Ni$_{80}$Fe$_{20}$) film is deposited on the substrate at a base pressure of $1.2 \times 10^{-7}$ torr, together with an Aluminum capping layer of 2 nm, to avoid fast oxidation. Finally, the sample is then placed in an Acetone bath for lift-off. The resulting structures consisted of nanomagnets with lengths $L = 360$ nm and widths $W = 120$ nm. The dimensions of nanomagnets are chosen, to ensure thermally-driven moment reorientations in the nanomagnets to occur at a timescale of a few seconds, perfectly matching temporal resolution of our PEEM imaging (10 s per image). The sample was left in a vacuum box at room temperature for over 4 weeks, to ensure annealing and relaxation towards low-energy configurations[2,3,12], before the sample was transferred to PEEM for imaging.

### Entropy determination
Since artificial spin ice typically only samples thousands of configurations from $2^N$ potential configurations and temperature resolution is limited, the sampling is inadequate to directly tabulate entropy. Previous studies of artificial spin ice have shown that the conditional entropy of local configurations can effectively upper bound experimental entropy[2,27]. The process is as follows: a subset of all spins, $\sigma_\Lambda$, is chosen to sample the entropy. Within that set, another subset, $\sigma_\Gamma$, is selected and will be used as the spins that "reveal" information and provide tighter bounds on the entropy. The conditional entropy, or the "surprise factor" of revealing new information from $\sigma_\Gamma$, is defined as

$$S(\sigma_\Lambda | \sigma_\Gamma) = - \sum_{\sigma_\Lambda, \sigma_\Gamma} P(\sigma_\Lambda, \sigma_\Gamma) \log_2 P(\sigma_\Lambda | \sigma_\Gamma). \tag{1}$$

The experimental spin configurations estimate all probabilities by counting: the probabilities of existing in each of the subset states, $P(\sigma_\Lambda, \sigma_\Gamma)$, the probability when the inner subset, $\sigma_\Gamma$, is in a specific state, $P(\sigma_\Gamma)$, and the conditional probability that the bigger subset state exists given the smaller subset state, $P(\sigma_\Lambda | \sigma_\Gamma)$. The conditional probability is calculable via Bayes' theorem, $P(\sigma_\Lambda | \sigma_\Gamma) = P(\sigma_\Lambda, \sigma_\Gamma)/P(\sigma_\Gamma)$. Information theory connects conditional entropy with an upper bound on entropy per spin. Specifically, the entropy per bit is upper bounded by the conditional entropy "revealed" by the additional spins in the larger subset divided by the number of additional spins. That is, for the correct $\sigma_\Lambda$ and $\sigma_\Gamma$, $n_\Gamma s \leq S(\sigma_\Lambda | \sigma_\Gamma)$ where $n_\Gamma$ is the number of spins in the inner subset. With this lattice, we elected to choose each vertex as sampling subsets and spin subsets, $\sigma_\Lambda$, and sub-subsets $\sigma_\Gamma$ shown in Fig. 3a. The sampling sets individual entropy densities are plotted in Fig. 3b as well as the average of all the entropy densities. Conditional probabilities and the resulting entropy estimations were computed from experimental spin configurations at all temperatures recorded.

### Configurational Entropy estimation
The entropy per bit due to the combinatorics of Type B vertex placement is calculated for the purpose of comparison with experimental entropy. The first and uppermost bound comes from the equivalent of a Pauling estimate: assume each unit (in this case, paired plaquettes as illustrated in Fig. 2e and Supplementary Fig. 2a) is uncoupled from its

neighbors and multiplicity of states are enumerated without consideration the state of other plaquettes. Two Type B vertices may occupy six possible sites on these set of two plaquettes. The multiplicity for two Type B vertices forced onto paired plaquettes with six possible sites is simply $\binom{6}{2}$, which is then multiplied by two in consideration of spin reversal symmetry. The bitwise entropy is then

$$s = \frac{1}{10}\log_2 2\binom{6}{2}, \qquad (2)$$

noting the division by the 10 spins that comprise paired plaquettes. This figures to $s \approx 0.4907$.

Neighboring sites will reduce this entropy by occasionally fixing the position of a Type B vertex, so a brute force estimate was conducted to further bound the entropy. Consider a finite-sized Apamea lattice with $N$ nanomagnets and $\Omega_B$ possible Type B configurations. To visualize the valid Type B configurations, one may consider a graph where each node is a pair of plaquettes, there are four edges that connect the nodes in a square lattice corresponding to shared Type B vertices, and two self-connections at each node to represent the potential Type B sites that are not shared (Supplementary Fig. 2b). All valid configurations select exactly two edges per node to contain a Type B vertex. The exact bitwise entropy we are trying to estimate is

$$s = \frac{1}{N}\log_2 2\Omega_B, \qquad (3)$$

the factor of 2 in the argument accounting for the spin flip symmetry associated with each Type B configuration. We assume that we may sample a random set of Type B configurations, $\omega_T$, and equate the ratio of valid configurations, $\omega_B$, those without overlapping Type B vertices, to the ratio of $\Omega_B$ to the total possible states, $\Omega_T$. That is,

$$\frac{\Omega_B}{\Omega_T} \approx \frac{\omega_B}{\omega_T} \qquad (4)$$

for a sufficiently large sample of the phase space. As the scale of these multiplicities were quite different for growing system size, the bitwise entropy was calculated numerically as follows:

$$s = \frac{1}{N}\left(1 + \log_2\Omega_T + \log_2\omega_B - \log_2\omega_T\right). \qquad (5)$$

Practically speaking, this was calculated by repeatedly placing a set of the correct number of Type B vertices on sites at random and recording the number of total attempted configurations, $\omega_T$, until $\omega_B = 100$. Sampling in this way was repeated several times at multiple system sizes so that the entropy's standard deviation of the mean was adequately small (Figure S2). The estimated entropy converges a little above 0.45, lower than the Pauling estimate.

### Ergodicity and the stress metric

The stress metric[29] is the difference between the time average and ensemble average of a system. In the context of a binary variable system this metric is defined as

$$\Omega(t) = \frac{1}{N}\sum_j \left(\frac{1}{t}\int_0^t S_j(s)ds - \frac{1}{N}\sum_r S_r(t)\right)^2. \qquad (6)$$

$N$ is the number of spins and $S_j(t)$ is the time-dependent Ising spin configuration. This metric decays to zero as the time and space averages become equivalent. We calculated the time evolution of $\Omega(t)$ at every temperature and fit its decay to a power law with power $z$.

## Data availability
The coordinate files of the nanomagnets generated in this study have been deposited in the Zenodo database under accession code 8250892.

## Code availability
The code used to analyze the data in this study is available upon request.

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

## Acknowledgements

The authors thank R.V. Chopdekar and A. Kleibert for their support and fruitful discussions. Part of this work was performed at the Advanced Light Source and the Molecular Foundry, Lawrence Berkeley National Lab, in addition to the SIM beamline of the Swiss Light Source. The Advanced Light Source and Molecular Foundry are supported by the Director, Office of Science, Office of Basic Energy Sciences of the Department of Energy under Contract No. DE-AC02-05CH11231. The work of F.C. and M.S. was carried out under the NNSA of the U.S., DoE at LANL, Contract No. DE-AC52-06NA25396 (LDRD grant - PRD20190195), LA-UR-22-29110. K.H. acknowledges support from the Swiss National Science Foundation (Project No. 200020_172774). C.N. was funded by DOE-LDRD grant 2017014ER.

## Author contributions

A.F. initiated the project, planned, and executed the experiments with support from K.H., S.D., and A.S.; M.S. analyzed and interpreted the data with support and input from F.C., A.F., and M.S. wrote the manuscript with input from all authors. C.N. and A.F. supervised the project.

## Competing interests

The authors declare no competing interests.
