## [Peer Review File · Nature Communications]

Real-space observation of ergodicity transitions in artificial spin iceREVIEWER COMMENTS

Reviewer #1 (Remarks to the Author):

This work investigates the timelessly interesting question of ergodicity -where time averages match ensemble averages- in the case of a vertex-frustrated artificial spin ice. Unfortunately, the system studied is not ideal for the purpose, the methods are not particularly well suited, and the results fall short of being noteworthy.

Artificial spin ices are not the best choice for the investigation of thermodynamic properties, equilibration, dynamics and ergodicity /ergodicity breaking. They are notoriously difficult to equilibrate, usually for causes external to the model system itself. Additionally, the number of components achieved in practice in samples is very far from the thermodynamic limit. Some of these shortcomings become apparent, for example, in the observed peculiar temperature dependence of the stress metric, related to fabrication details and possible finite size effects of the sample rather than generic spin ice features.

One of the claims in the abstract is that this work serves as evidence that real-space imaging is a useful tool to explore fundamental laws of thermodynamics. This claim is not substantiated. The low number of observations results in the need for indirect methods to estimate quantities (the heading "direct entropy determination" is misleading). The possible advantage of the experimental method chosen, real-space observation, is unfortunately ultimately marred by the need for estimates and calculations.

Finally, it is not clear how the purported ability to directly visualise transitions, even if true, is a significant result. This might well be the case, but it is not shown or discussed properly in the article.

In conclusion, while the work presents an interesting experimental and theoretical effort towards the development of new techniques to explore thermodynamics and dynamics of condensed matter systems, I do not think this work is suited for publication in Nature Communications. With some additional discussion of the results it might be suited for a more specialized journal.

Reviewer #2 (Remarks to the Author):

The work by M. Saccone et al. discusses interesting aspects of dynamics in artificial spin ice (ASI) systems, relating to ergodicity and the impact constraints of it have on the observed order and dynamics.

A list of comments/feedback from my side includes the following:

- * Line 100-104: The authors discuss kinetic limitations for the three-nanomagnet vertices. If that is the case, how does the following sentence play strongly in, since the system might be locked in disordered states?
- * The term “plaquette” is being used but has not been properly introduced when discussing the lattice. Which parts of the lattice is this actually referring to?
- * The discussion of the direct entropy determination, is not easy to follow and even confusing. The authors should amend and use graphical ways to explain how exactly the theoretical thresholds are calculated. The lines explaining the tiling of Type B vertices on the lattice units are very hard to follow as the terminology used has not been properly introduced (mostly with respect to the lattice). I honestly do not know how to exactly interpret Supplementary Figure 2a. Some more input to the reader is needed on all of these.
- * Line 140-141. Fluctuation frequencies are mentioned here, which I guess originate from PEEM-XMCD imaging. How are these actually calculated? What is the error bar on the reported values? Is the difference statistically significant as they do not differ too much?
- * Line 169-171, “Only by cooling.... to reach a new ensemble.”: The statements in that sentence are somewhat contradicting. The authors should explain more carefully what they mean here. Generally cooling should inhibit fluctuations.... (Probably relates to some of the point in the bottom of this list)
- * How exactly have the exponents for the decay of the stress metric been extracted? Which parts of Fig. 4a have been fitted for this and why? How exactly have the uncertainties reported in Fig. 4b been calculated, given the data of Fig. 4a. The authors might not think that this is worth discussing in detail, but from a first look on Fig. 4 is it not clear how that data has been treated to reach the conclusions stated in the text.

* Could not help wondering what the effect of the temperature dependence of the interaction strength might be and how it could couple to the observations of Fig. 4? As temperature rises not only the fluctuation rate increases, but as these nanomagnets are far from ideal point dipoles, their magnetization expectation value (within the observation window) should also reduce, resulting in a change in interaction. The latter could further be influenced by the coordination of the vertex (2-, 3- or 4-fold) which might help in counteracting some of temperature effect, depending further on the vertex type (I, II,.... A,B,... etc). Could it be that the claim from the title of observations of transitions driven by ergodicity has more to do with that interaction modification?

* I think the authors try to make the point (but not as clear as it could be) that ergodicity is being partially 'restored' while cooling the sample down. Even though this might be counterintuitive at first, it is worth discussing it in more detail and look for the source of it. Figs. 4c-e contain some data that could help in that direction. According to these it seems that 3-fold vertices get activated in the intermediate temperature range. What is the main reason for this? 3-fold vertices occupy a variety of positions in the lattice and neighbour 4-fold, 3-fold and 2-fold vertices. It could be of use to investigate how the energy levels for all vertex coordinations and types relate to each other and what the potential routes towards ground state ordering might be. The entropy estimates are strong indicators but do not fully resolve the issue. Looking at the plot provided by the authors (Fig. 3b) something seems to be "leaking" (entropy-wise) from 3-fold to 4-fold vertices in the intermediate temperature, but the question is how and why? A better understanding of this could be potentially very useful for the control and design of ergodicity in future lattices, employing proper geometry and materials. This could open interesting new vistas on the dynamics and kinetics for ASIs.

* The discussion section is very short and focusing on subjective projections of the work presented. Some more reflections and deeper analysis of the observations presented needs to be done here, putting weight on the actual novelty and importance of the results, backed with objective solid arguments. Maybe it could be used to bring up the discussion on the previous two points of my comment list.

* The manuscript generally suffers in a lot of places from syntax and grammatical errors, which should be straight forward and easy to amend.

* I would like to encourage the authors to go through all the figure captions and make sure that these are explain in full detail everything in the figures. Improvements in the graphical language used in the figures themselves could also enhance the communications (see some points above).

Thoughts on the topics presented in this manuscript have been in the center of attention of the ASI community since its early days and have been investigated mostly theoretically, but also to some extend experimentally. Even though the authors mention this as well, albeit briefly, they are neither referencing properly past works or even attempting relating their findings to them.

A brief list of some works that I can bring up (more exist...), are:

- * PRL 105, 017201 (2010): Modelling of population dynamics & some array finite size effects.
- * PRL 109, 037203 (2012): Disorder and ground state interplay.
- * PRB 95, 104022 (2017): Relaxation probed by XPCS.
- * Scientific Reports 6, 37097 (2016): Temperature dependent relaxation rates probed by SQUID magnetometry.
- * Nature Physics 14, 375 (2018): Evolution of an ASI system in time and temperature in terms of its autocorrelation (Fig. 3d and related discussion).
- * New Journal of Physics 23, 033024 (2021)

I am not proposing adding references in bulk here, but some more detailed discussion needs to be done to interface better to the work that has been already done and highlight new views and inputs from the present manuscript.

Concluding my report, I do not support the publication of the manuscript in Nature Communications in its current state. Even though the authors have - in my opinion - an interesting system and effect under their scope, the presentation and more crucially the discussion do not pay court to them.

We thank both Reviewers for their reviews, which lead to an improvement of our manuscript, overall. Below you can find our point-by-point responses to the Reviewer reports. Text based changes to the manuscript are highlighted in red font

Reviewer #1 (Remarks to the Author):

This work investigates the timelessly interesting question of ergodicity -where time averages match ensemble averages- in the case of a vertex-frustrated artificial spin ice. Unfortunately, the system studied is not ideal for the purpose, the methods are not particularly well suited, and the results fall short of being noteworthy.

Artificial spin ices are not the best choice for the investigation of thermodynamic properties, equilibration, dynamics and ergodicity /ergodicity breaking. They are notoriously difficult to equilibrate, usually for causes external to the model system itself. Additionally, the number of components achieved in practice in samples is very far from the thermodynamic limit. Some of these shortcomings become apparent, for example, in the observed peculiar temperature dependence of the stress metric, related to fabrication details and possible finite size effects of the sample rather than generic spin ice features.

Response: We respectfully disagree. While the viewing area of characterization is limited, each array involves 28,500 magnets, which would make it well into the thermodynamic limit. Moreover, finite sizes in general would makes equilibration easier rather than harder. (It has been shown in various published works that a small size artificial spin ice would reach its ground state even with notoriously poor AC demagnetization protocols.)

Moreover, while personal opinions can vary, artificial spin ices have factually and repeatedly been shown to be attractive model systems to directly image the thermodynamics of frustrated magnetism, both in equilibrium and out-of-equilibrium. In too many works to report here, statistics of states have aligned surprisingly well with equilibrium states predicted by Monte Carlo simulations. We will just mention ground state observations via thermal annealing (Zhang et al. Nature 2013, Farhan et al. PRL 2013, Farhan et al. Nature Comms. 2016), direct and quantitative imaging of emergent magnetic monopole dynamics (Farhan et al Science Advances 2019), the observation of equilibrium phase transitions (Hofhuis et al. 2022), of kinetic slowing-down at transitions (*Nature communications*, 2015, 6.1: 8278).

In fact, on the contrary, it is surprising how well these systems have been aligning to simple Monte Carlo predictions at equilibrium.

What the reviewer claims was certainly true before 2011-2012, that is, before the introduction of superparamagnetic versions of artificial spin ice, or of protocols of thermal annealing. Since then, it has been demonstrated that thermal equilibrium can be reached.

One of the claims in the abstract is that this work serves as evidence that real-space imaging is a useful tool to explore fundamental laws of thermodynamics. This claim is not substantiated. The low number of observations results in the need for indirect methods to estimate quantities (the

heading "direct entropy determination" is misleading). The possible advantage of the experimental method chosen, real-space observation, is unfortunately ultimately marred by the need for estimates and calculations.

Response: The Reviewer is certainly correct that entropy extraction via calorimetry in these systems is practically impossible. However, extracting entropy is not the scope of this work. Conditional entropies, based on conditional probabilities, are used not as thermal observables, but as quantifiers of different kinetics for different degrees of freedom.

To avoid confusion, we have altered the text to include:

“To do so, following ref [24], we consider subsets of the lattice, coordination two, three, and four island vertices (Figure 3a), and ask the question ‘how much information is hidden by only observing a part of this subset?’ If the rest of the microstate is determined by only the part of the subset with perfect probability, the entropy per bit is zero, but if the rest of the subset is entirely random, the entropy per bit is one. The details of the intermediate probabilities are determined by Bayes’ theorem and basic information theory^{2,24} (see Methods). Calculating the entropy for each temperature and coordination of vertex, as well as the appropriate weighted average to determine the total entropy, we see that the entropy bound generally decreases with decreasing temperature as expected (Figure 3b), but with a stark jump downward between 300 and 290 K for the coordination two and four vertices while the bound from the coordination three vertices remains nearly constant.”

Finally, it is not clear how the purported ability to directly visualise transitions, even if true, is a significant result. This might well be the case, but it is not shown or discussed properly in the article. In conclusion, while the work presents an interesting experimental and theoretical effort towards the development of new techniques to explore thermodynamics and dynamics of condensed matter systems, I do not think this work is suited for publication in Nature Communications. With some additional discussion of the results it might be suited for a more specialized journal.

Response: The value of a real-space, real-time characterization of the elementary degrees of freedom can be seen in our work in the possibility to extract stress metrics that average on the individual moments, something generally impossible in natural materials, but possible in simulations. This allows for a microscopic extraction of differences between averages over space and over time, which goes at the heart of the problem of ergodicity.

The notion of ergodicity belongs to statistical mechanics, an approach which ties macroscopic thermodynamic observables to the microscopic degrees of freedom.

Nonetheless, we do agree with the Reviewer that more could have been done to exploit the access to a microscopic picture. Thus, we have added new data in two new figures, and their discussion. There we look at microscopic flip rates of individual moments, and transition rates among vertex configurations, and discussed them in the context of the results from the extraction of the stress metrics.

We thank the reviewer for the effort and for helping us to improve the manuscript.

Reviewer #2 (Remarks to the Author):

The work by M. Saccone et al. discusses interesting aspects of dynamics in artificial spin ice (ASI) systems, relating to ergodicity and the impact constraints of it have on the observed order and dynamics.

A list of comments/feedback from my side includes the following:

** Line 100-104: The authors discuss kinetic limitations for the three-nanomagnet vertices. If that is the case, how does the following sentence play strongly in, since the system might be locked in disordered states?*

Response: We argue that the limited kinetics can create disordered states, specifically ones that defy ergodicity. Local minima are possible, but the system does not globally minimize the number of Type B vertices as referred to in lines 93-95. The context of this statement is a bit confusing, so we revise the text accordingly:

“In other words, the system is stuck in a low-energy configuration that does not pass through higher energy states to find the ground state, for both charge- and moment degrees of freedom (see Figure 2b).”

** The term “plaquette” is being used but has not been properly introduced when discussing the lattice. Which parts of the lattice is this actually referring to?*

Response: We now define the term:

“Closing their flux around square and irregular hexagonal polygons formed by the nanomagnets, hereafter referred to as a plaquettes...”

** The discussion of the direct entropy determination, is not easy to follow and even confusing. The authors should amend and use graphical ways to explain how exactly the theoretical thresholds are calculated. The lines explaining the tiling of Type B vertices on the lattice units are very hard to follow as the terminology used has not been properly introduced (mostly with respect to the lattice). I honestly do not know how to exactly interpret Supplementary Figure 2a. Some more input to the reader is needed on all of these.*

Response: We appreciate the call to clarify these sections and have expanded the text accordingly. Supplementary Figure 2a is now alluded to more strongly by the main text to explain the calculations.

** Line 140-141. Fluctuation frequencies are mentioned here, which I guess originate from*

PEEM-XMCD imaging. How are these actually calculated? What is the error bar on the reported values? Is the difference statistically significant as they do not differ too much?

Response: To expand on this concept, we have added a new Figure 5 tabulating the transition rates between different vertex types to explain the kinetics of the system. This figure provides error bars and more obvious interpretations of the rates. Error bars are the standard deviation of the mean when averaging the rates over four separate time intervals.

** Line 169-171, "Only by cooling.... to reach a new ensemble.": The statements in that sentence are somewhat contradicting. The authors should explain more carefully what they mean here. Generally cooling should inhibit fluctuations.... (Probably relates to some of the point in the bottom of this list)*

Response: We agree that this is not an immediately obvious result. Further commentary is needed and has been added:

"By reaching an intermediate temperature, 270-290 K, the system fluctuates towards a new equilibrium ensemble. Since thermodynamic systems are forced towards free energy minima, and lowering the temperature shifts them, the system evolves anew to follow a new minimum. Lower temperature targets favor lower energy above multiplicity of states, pressing the Apamea lattice to seek long range order despite the kinetic barriers. This behavior hides in the fast relaxation of traditional materials, but the invisible is made visible in artificial nanomagnets."

** How exactly have the exponents for the decay of the stress metric been extracted? Which parts of Fig. 4a have been fitted for this and why? How exactly have the uncertainties reported in Fig. 4b been calculated, given the data of Fig. 4a. The authors might not think that this is worth discussing in detail, but from a first look on Fig. 4 is it not clear how that data has been treated to reach the conclusions stated in the text.*

Response: Explicit mention of our methodology has been added.

** Could not help wondering what the effect of the temperature dependence of the interaction strength might be and how it could couple to the observations of Fig. 4? As temperature rises not only the fluctuation rate increases, but as these nanomagnets are far from ideal point dipoles, their magnetization expectation value (within the observation window) should also reduce, resulting in a change in interaction. The latter could further be influenced by the coordination of the vertex (2-, 3- or 4-fold) which might help in counteracting some of temperature effect, depending further on the vertex type (I, II, A,B, ... etc). Could it be that the claim from the title of observations of transitions driven by ergodicity has more to do with that interaction modification?*

Response: The change in saturation magnetization of permalloy thin films does reduce as temperature increase as seen in a previous study with similar magnetic material (Saccone, Michael, et al. "Direct observation of a dynamical glass transition in a nanomagnetic artificial Hopfield network." *Nature Physics* 18.5 (2022): 517-521, Supplementary Figure 4a). Comparing to the 2.5 nm thick samples, the magnetization reduces from 810 kA/m to 750 kA/m over our current experimental window of 250-310 K. Dipolar interaction energy is proportional to magnetization squared, which decreases by 14.3% over this range if all other factors are held constant. Because the higher temperature ensembles would move closer to paramagnetism when interaction strength is lowered, this translates to a larger range of effective temperatures rather than a smaller one. Additional details of the exact magnetic textures evolving over temperature could occur, but probably would not contribute to the interactions more than 20%. This itself could be the subject of another paper as it is an effect not commonly addressed throughout the field. However, this does not alter our interpretation of the data.

** I think the authors try to make the point (but not as clear as it could be) that ergodicity is being partially 'restored' while cooling the sample down. Even though this might be counterintuitive at first, it is worth discussing it in more detail and look for the source of it. Figs. 4c-e contain some data that could help in that direction. According to these it seems that 3-fold vertices get activated in the intermediate temperature range. What is the main reason for this? 3-fold vertices occupy a variety of positions in the lattice and neighbour 4-fold, 3-fold and 2-fold vertices. It could be of use to investigate how the energy levels for all vertex coordinations and types relate to each other and what the potential routes towards ground state ordering might be. The entropy estimates are strong indicators but do not fully resolve the issue. Looking at the plot provided by the authors (Fig. 3b) something seems to be "leaking" (entropy-wise) from 3-fold to 4-fold vertices in the intermediate temperature, but the question is how and why? A better understanding of this could be potentially very useful for the control and design of ergodicity in future lattices, employing proper geometry and materials. This could open interesting new vistas on the dynamics and kinetics for ASIs.*

Response: We entirely agree. Two new figures have been added to evaluate the transition rates between vertex types and visualize the time evolution in weak ergodicity breaking and frozen regimes respectively. We hope the thoroughly expanded analysis adequately addresses the reviewer's curiosity.

** The discussion section is very short and focusing on subjective projections of the work presented. Some more reflections and deeper analysis of the observations presented needs to be done here, putting weight on the actual novelty and importance of the results, backed with objective solid arguments. Maybe it could be used to bring up the discussion on the previous two points of my comment list.*

Response: We expanded the Discussion section, following the Referee's advice and added more context linked to the Referee's suggested references.

** The manuscript generally suffers in a lot of places from syntax and grammatical errors, which should be straight forward and easy to amend.*

Response: Further copy editing has been applied throughout.

** I would like to encourage the authors to go through all the figure captions and make sure that these are explain in full detail everything in the figures. Improvements in the graphical language used in the figures themselves could also enhance the communications (see some points above).*

Thoughts on the topics presented in this manuscript have been in the center of attention of the ASI community since its early days and have been investigated mostly theoretically, but also to some extend experimentally. Even though the authors mention this as well, albeit briefly, they are neither referencing properly past works or even attempting relating their findings to them.

A brief list of some works that I can bring up (more exist...), are:

** PRL 105, 017201 (2010): Modelling of population dynamics & some array finite size effects.*

** PRL 109, 037203 (2012): Disorder and ground state interplay.*

** PRB 95, 104022 (2017): Relaxation probed by XPCS.*

** Scientific Reports 6, 37097 (2016): Temperature dependent relaxation rates probed by SQUID magnetometry.*

** Nature Physics 14, 375 (2018): Evolution of an ASI system in time and temperature in terms of its autocorrelation (Fig. 3d and related discussion).*

** New Journal of Physics 23, 033024 (2021)*

I am not proposing adding references in bulk here, but some more detailed discussion needs to be done to interface better to the work that has been already done and highlight new views and inputs from the present manuscript.

Concluding my report, I do not support the publication of the manuscript in Nature Communications in its current state. Even though the authors have - in my opinion - an interesting system and effect under their scope, the presentation and more crucially the discussion do not pay court to them.

Response: We thank the Reviewer for these suggestions. We have added more context in light of these citations and added them in the final part of the Discussion section.

REVIEWERS' COMMENTS

Reviewer #1 (Remarks to the Author):

The work presents interesting methods and represents an advance in the area of artificial spin-ice systems, a fertile and productive area that in many cases has served as a platform for experiments that are impossible to realise in other systems.

The primary issue with these findings lies in the lack of a persuasive argument from the authors to establish a correspondence between the observed behaviour and the equilibrium dynamics of a system in the thermodynamic limit. This greatly undermines the relevance of the results.

The work is better suited for a more specialised journal.

Reviewer #2 (Remarks to the Author):

I would like to raise some remarks, which primarily concern the reply to Referee 1 and might be of use for consideration for this and other journals that host such works. The thermodynamic limit relates explicitly to particle populations approaching Avogadro's number or larger. Systems discussed by the authors are not meeting this conditions. Having said that, I do not strongly disagree with the authors, but it is important in the scientific community to have a clear consensus about the meaning of words, which I think is what the Reviewer 1 was also after with his suggestion. If that is not the case, the communication of scientific ideas, arguments and observations is going to be very difficult.